# Assessing the Effect of Data Quality on Distance Estimation in Smartphone-Based Outdoor 6MWT

**DOI:** 10.3390/s24082632

**Published:** 2024-04-20

**Authors:** Sara Caramaschi, Carl Magnus Olsson, Elizabeth Orchard, Jackson Molloy, Dario Salvi

**Affiliations:** 1Department of Computer Science and Media Technology, Internet of Things and People Research Center, Malmö University, 21119 Malmö, Sweden; carl.magnus.olsson@mau.se (C.M.O.); dario.salvi@mau.se (D.S.); 2Oxford University Hospitals NHS Foundation Trust, Oxford OX3 7JX, UK; elizabeth.orchard@ouh.nhs.uk (E.O.); jackson.molloy@ouh.nhs.uk (J.M.)

**Keywords:** 6MWT, distance estimation, data reliability, physical assessment

## Abstract

As a result of technological advancements, functional capacity assessments, such as the 6-minute walk test, can be performed remotely, at home and in the community. Current studies, however, tend to overlook the crucial aspect of data quality, often limiting their focus to idealised scenarios. Challenging conditions may arise when performing a test given the risk of collecting poor-quality GNSS signal, which can undermine the reliability of the results. This work shows the impact of applying filtering rules to avoid noisy samples in common algorithms that compute the walked distance from positioning data. Then, based on signal features, we assess the reliability of the distance estimation using logistic regression from the following two perspectives: error-based analysis, which relates to the estimated distance error, and user-based analysis, which distinguishes conventional from unconventional tests based on users’ previous annotations. We highlight the impact of features associated with walked path irregularity and direction changes to establish data quality. We evaluate features within a binary classification task and reach an F1-score of 0.93 and an area under the curve of 0.97 for the user-based classification. Identifying unreliable tests is helpful to clinicians, who receive the recorded test results accompanied by quality assessments, and to patients, who can be given the opportunity to repeat tests classified as not following the instructions.

## 1. Introduction

Standard physical activity tests, such as the Queens College Step Test [1], the Timed Up and Go Test (TUG) [2], or the 6-Minute Walk Test (6MWT), are often employed to assess physical capacity and functional performance on populations with limited mobility or cardiorespiratory and peripheral vascular disease.

Within this study, we focus on the 6MWT, which is a clinical test widely used by physicians to monitor the progress and deterioration of a patient’s physical and functional capacity [3]. The distance walked during a 6MWT can be a relevant indicator of health status within an elderly population [4], and among others, it is helpful in a wide range of conditions, for example, within cardiovascular and respiratory diseases [5,6].

Performing conventional, in-clinic 6MWT requires patients to physically travel to the hospital and specialized clinical staff to monitor the test, resulting in a burdensome practice in terms of time and costs. Additionally, it requires environmental conditions, such as an obstacle-free hallway, preferably at least 30 m long, which is not always accessible. Thanks to technological advances in sensors and mobile devices these tests can be performed remotely and in patients’ homes or communities [7]. Such remote 6MWTs reduces the required effort for patients and clinical staff, which promotes more frequent and, therefore, more robust monitoring of patients [8]. Several algorithmic approaches have been proposed to estimate the 6-Minute Walk Distance (6MWD) from raw sensor data. Outdoor solutions are commonly based on Global Navigation Satellite System (GNSS) positioning while indoor solutions typically depend on inertial sensors or other modalities, such as cameras [9].

The primary purpose of GNSS is to offer precise and real-time global navigation and positioning within a terrestrial reference frame. However, while its use in smartphone applications has proven to be generally reliable, it can occasionally be susceptible to inaccuracies [10]. Specifically, within the context of the 6MWT, environmental conditions, handling of smartphones, and low quality of embedded antennas may significantly impact positioning performances [11,12]; thus, research developments must take these aspects into account, especially within the healthcare context.

In this paper, we focus on outdoor GNSS- and smartphone-based 6MWT. We compare the algorithms that compute the 6MWD from positioning data. Within this context, we highlight the importance of using filtering rules across the different algorithms.In the context of data quality assessment, we investigate the role of signal features and the related classification algorithms coefficients, which enable the differentiation of the tests according to quality considerations.

The article is structured as follows: Section 2 introduces related research within the field, while Section 3 covers the datasets’ description and considered methods. Section 4 shows the obtained results, which are then discussed in Section 5. Finally, Section 6 concludes the article by summarizing the main findings and outlines suggested future work.

## 2. Related Work

The measure of the walked distance can be approached using different types of technology according to the context of data collection and availability. Within the pedestrian navigation field, outdoor environments allow data fusion between Inertial Measurements Unit (IMU) and positioning data to track a person’s walking activity. For example, Basso et al. [13] proposed a real-time pedestrian navigation system by placing the smartphone inside a pocket making use of both GNSS and inertial data. Jian et al. [14] made use of Pedestrian Dead Reckoning (PDR) and GNSS information by using factor-graph optimization. However, this implementation relies on considering multiple smartphones within the area of interest to triangulate information. They furthermore explore the use of pseudo-range measurements, which correspond to the distance between each satellite and the user’s antenna [15].

Also within the context of the 6MWT, studies propose using smartphones IMUs [9,16]. For example, Mak et al. [8] conducted an extensive study concerning in-clinic and home-based 6MWT using inertial sensors embedded in iPhone 7 smartphones, Apple.Their findings are promising, but rely only on the step count, as measuring distance from IMU is not easy to achieve reliably.

To measure distance accurately, it is more common to rely on the GNSS signal within outdoor contexts. Regardless of the noise in the GNSS signal, the validity and reproducibility of the outdoor 6MWT have been proven [17].

One such example is Salvi et al. [18], where the authors developed an algorithm to compute the 6MWD for the outdoor scenario. Their algorithm works by downsampling the GNSS signal by selectively using the positions with the highest accuracy, as computed by the operating system of the phone, over a time window of 5 s and summing the distance between each selected position. From lab experiments, the authors reported a mean difference between the output of the algorithm and the ground truth of −0.80 m ± 18.56 m standard deviation for the outdoor scenario. Tests performed later with 30 pulmonary hypertension patients during 6 months using supervised in-clinic tests and unsupervised outdoor tests showed that the outdoor algorithm was also valid and repeatable [19].

Ata et al. [20] developed the VascTrac iPhone application [21] developed by O. Aalami and available on the iOS app store. The authors tested the app on a peripheral arterial disease population, where they considered a digital version of the 6MWT provided from the iPhone’s distance and step-counting algorithms. Their evaluation shows that the distance algorithm results in an overestimation of the reference measure, with the initial relative mean and standard deviation of 43%±42% and after using a correction factor reduced to 8%±32%. Additionally, they tested multiple phone positions such as in the hand, pocket, and bag or purse. They conclude that best performances arise when the phone is hand-held while reproducing the test.

Research developed by Ziegl et al. [22] investigated how the device positioning and filtering techniques can impact the walking distance computation. To test this, they collected data from 166 recordings using a mobile phone application, and based on this proposed an algorithm based on Kalman filter operations obtaining a 3.7% mean error.

Ogris et al. [23] extended the analysis of the Kalman-based algorithm by considering multiple combinations of filters and convolutions, and reached a promising performance where the relative percentage error of 23 experiments out of 24 were below the chosen threshold of 5%. However, in both studies, the collected tests were performed by healthy participants. This subsequently leaves some uncertainty if the technique generalises well on patients.

Van Oirschot et al. [24] implemented a smartphone-based 2-minute walk test based on Global Positioning System (GPS) data using a proprietary algorithm, which reconstructs the walked path. They validated the reliability of their method with 25 persons with relapsing-remitting Multiple Sclerosis (MS) and 79 healthy controls and compared the computed distance with reference markers. The difference between computed distance and reference was 5% on average, while the details of the employed algorithms are not published, and are thus hard to build further research on, the paper provides a relevant filtering criterion for the test. If the time difference between the first and last GPS data point (i.e., the overall duration) is not within 120±20 s the test is automatically discarded. This is reasonable given the frequency of GPS frequency between 1 and 2 Hz. However, for a longer test duration, it is harder to fix an established time margin.

As also addressed by Van Oirschot et al. [24], the challenge of dealing with low-quality positioning data is a problem to consider carefully. Particularly environments, such as ”urban canyons” are known to affect positioning from satellites [25]. Paziewski [10] elaborates on factors that affect the GNSS accuracy in smartphone applications for precise localization purposes. In relation to the 6MWT, Stienen et al. [11] highlights the difficulties incurred when computing the 6MWD in a smartphone app for remote 6MWT, including indoor environments, being near high buildings, in rectangular-shaped paths, and at low speeds. Accuracy and quality issues in GNSS data and their relationship with the 6MWT have thus been recognized by previous research.

However, in real-life scenarios, it would be ideal to identify low-quality data during the data collection phase to ensure that recorded tests with poor data quality are not considered by healthcare professionals. Therefore, there is still the need to perform the following: (1) better analyse how low quality GNSS data affects the estimation of the 6MWD, and, importantly, (2) explore if it is possible to automatically characterise low-quality recordings by, for example, analyzing the features extracted from the signal of a smartphone-based 6MWT.

Informed by the above work, our study relies on walking tests performed outdoors by recording the GNSS signal and with participants holding the phone in their hand during the test. Given the need for realistic settings, we could not assume access to multiple phones for triangulation to improve positioning. Technical issues that affected some participants during data collection also meant that inertial measurements are not available for all tests. For the purposes of this paper, we are therefore focusing the analysis on distance estimation methods that rely only on GNSS information, even if we acknowledge the potential for data fusion techniques to further improve results in future studies.

## 3. Materials and Methods

In this study, we use an existing dataset of positioning traces collected with a smartphone during a 6MWT. Approximately half of the used dataset was collected in a way that introduced noise on purpose, to ensure that our data contained erroneously conducted tests. An overview of our research is shown in Figure 1.

The traces were first pre-processed to remove positions deemed inaccurate and to obtain a constant sampling rate. The data were then used in a series of algorithms for computation of the 6MWD. The output of those algorithms was compared with the ground truth measurement to extract statistics about accuracy. We additionally implemented classifiers to distinguish between traces where the distance estimates are more accurate and those that produce inaccurate results (error > 30 m). We also classify traces where the user walked following a regular path (classified as conventional) and those where the user did not follow the instructions for how to conduct a 6MWT (classified as unconventional).

### 3.1. Dataset

The data used in this article were collected from outdoor 6MWTs. All tests were performed using the smartphone app named Timed Walk, version 0.3.2, developed by D. Salvi [26]. This app is based on previous work [18,19] and runs on both Android and iOS-based smartphones. It collects GNSS signals from the geolocation web API together with inertial measurements such as the triaxial accelerometer and gyroscope data through the motion and orientation APIs, and computes the walked distance using the algorithm described in [18]. All tests were performed with the smartphone in one hand and a a distance measuring wheel (trundle wheel or odometer, Qingdao Qingqing Hardware Tools Co., Ltd., Qingdao, China) in the other hand to measure the reference walked distance.

Based on research conducted in lab tests [18], it is known that the distance estimation of the Timed Walk app is more reliable when the walked path is straight, and if the environment does not block or reflect satellite signals. Thus, the smartphone app presents instructions to the user before the start of the test, to perform the test reliably. These include walking in a straight line or with a gently curved path, ideally in weather conditions where the sky is visible and avoiding areas with tall buildings or walking under dense trees. Due to a standard measure for app privacy used by mobile phone operating systems, users were instructed to keep the display switched on and to avoid using other apps during the test. Otherwise, if the app is placed in the background (e.g., when the user receives a phone call) or if the screen is switched off, the app will stop receiving position updates.

The collected data come from two different sets. The first one consists of 107 recordings collected by 10 volunteers recruited at the Oxford University Hospitals NHS Foundation Trust who were asked to perform 10 or more outdoor 6MWTs. Of these, 7 were cardiac patients under treatment at the clinic and 3 were healthy volunteers. Data were collected within a study with ethical approval from the UK National Health Service Health Research Authority (protocol reference numbers: 17/WM/0355) and informed consent was obtained from all participants involved. Each volunteer was asked to perform five conventional tests following the app instructions, and five “unconventional” tests, in which they would, on purpose, not follow the instructions, for example, by including several sharp turns. After each test, the participant noted if the test was made conventionally or unconventionally. In Figure 2, two examples of conventional and unconventional tests are shown.

An additional 62 recordings were collected at Malmö University by four healthy researchers. These recordings were acquired following different types of paths, namely, “regular”, which aligns with the app instructions, “stop and go”, which consists of intervals of alternating straight walking and standing still, “back and forth”, which consists of walking back and forth a 20 m distance, and four different types of circular paths named “circles0”, “circles1”, “circles2”, and “circles3” characterised by a decreasing radius of the circular path. Recordings of the type “regular” and “stop and go” are both considered as “conventional” as the patient is walking straight, while “back and forth”, and all circular paths are considered as “unconventional”. The goal of these tests was to explore how distance estimation is affected by the shape of the walked path. These tests were performed with different durations, from 3 to 6 min, as this interval still ensured that sufficient positions were collected to be useful when analyzing data quality.

A single test recording consists of a time series of GNSS updates, each one including latitude, longitude, and elevation, together with the timestamp to which the recorded data corresponds. Additionally, each sample is associated with a measurement of the confidence interval of the position, i.e., the estimated horizontal accuracy radius in meters, computed by the operating system, the heading, i.e., the direction towards which the device is facing in degrees from the magnetic north, and the travelling speed in the current direction. The positional signal has a sampling frequency of approximately 1 Hz on all phones, with some jitter depending on conditions. In addition to the GNSS data, the smartphone’s step count, triaxial acceleration, triaxial rotation rate, and triaxial orientation were collected at a sampling frequency of 60 Hz. Inertial data were not available on all recorded traces due to a bug in the app that prevented inertial measurement collection on iPhones.

### 3.2. Data Pre-Processing and Filtering

A number of filters were implemented to remove samples that are likely to be affected by noise that would add error to the overall estimation. Filters were implemented as simple rules, mostly applied on a single or two consecutive samples. If any of the following was true, the sample was excluded from the computation of the distance:If the sample does not include a value for the altitude. This happens when the position is not computed using satellites, as mobile operating systems also make use of cellular network triangulation or WiFi networks to estimate position, while useful in many use cases, these techniques give less accurate positions than GNSS and were thus excluded;If the confidence interval estimated by the operating system was above 25 m. This threshold was chosen as a compromise between accuracy (the lower the confidence interval, the more accurate the measurement is) and availability of samples (a higher confidence interval implies that fewer samples are discarded). Related to this, the app would only allow the test to start after at least one sample is received with a confidence interval below 15 m, to ensure that the GNSS system has achieved a strong “lock” on the signal transmitted by satellites. For each additional satellite being included in the signal, the confidence interval becomes reduced. A compromise between desired accuracy and waiting time was chosen as 15 m;If the average acceleration magnitude is below 0.5 m/s^2^. This indicates that the user could be still. Thus, if the sample is included, even small errors in the position would be added to the computation of the distance, even if no additional distance was actually walked. The threshold was chosen empirically from the data collected by the researchers, in particular, informed by the “stop and go” recordings. The value was selected so that the filtering is conservative and excludes only samples that correspond to non-walking segments;If the difference in steps between two consecutive points is zero. This indicates that there is no movement, and follows the same logic as the case above;If the time difference between two consecutive points is less or equal to zero. This can happen when the system occasionally returns old values, particularly when the visibility of the satellites is lost, and the operating system resorts to sending the last known location;If the speed computed between two consecutive points is above 5 m/s. This threshold was chosen as it is substantially higher than a human walking speed [27], very unlikely to happen in a 6MWT.

Figure 3 shows an example of a 6MWT GNSS signal without the application of any filter. Additionally, GNSS signals were resampled at 1 Hz in order to be consistent with algorithms that are designed to be used at a constant sampling frequency.

### 3.3. Algorithms for Walked Distance Estimation

We implemented and compared the following six algorithms: Alpha-Beta [28], Kalman 1D [29], Kalman 2D [30], Kalman smoothing [22], Quality-based Spatial Subsampling (QSS) [18], and a simple baseline algorithm that sums the distance between samples. The following four algorithms are based on the Kalman filter: Alpha-Beta, Kalman 1D, Kalman 2D, and Kalman smoothing. Kalman filter approaches have already been proven to be reliable in distance computation and position estimation [31,32]. The approach is based on a set of equations, which recursively estimate the state of a dynamic process to minimize the squared error mean [33]. For the algorithms that rely on Kalman filters, we used the resampled GNSS signal at 1 Hz to have a constant sampling frequency. This was performed to simplify the mathematical modelling needed in these filters and was also as suggested in some previous work [22].

**Baseline algorithm:** The simplest of the algorithms adds all the distances “as the crow flies” between each retrieved position, after initial pre-processing. Given that the GNSS signal can be noisy, integrating over time has the effect of adding up the error from each sample. This is exacerbated by the fact that, at the walking speed and at typical GNSS sampling frequency (1 Hz), recordings can contain several points located close to each other. This approach therefore tends to compute inaccurate distances unless particular care is taken to reduce noise. While not expected to perform as well as other approaches, including this algorithm is useful as an indication of how well a baseline approach stands up to more sophisticated algorithms.

**Quality-based sub-sampling (QSS):** This algorithm is described in [18]. The original paper did not provide a name for it, but for convenience, we will refer to it as quality-based sub-sampling (QSS). The algorithm integrates the distances between positions, as in the baseline algorithm, but sub-sampling is performed first using some heuristics, while receiving the position updates, the algorithm periodically (every 5 s) selects the position with the lowest confidence interval in a time window. For the purpose of our study, thresholds and periods were chosen to be the same as in the original version. The algorithm applies simple filtering rules similar to the ones that we applied in the data pre-processing phase, detailed in Section 3.2. In particular, it considers the time difference between two consecutive samples (which must be greater than zero), the maximum speed allowed (set to 2 m/s), and if the user is moving by checking the difference in step counting of consecutive samples.

**Alpha-beta:** This algorithm is based on the alpha-beta equations [34,35] and consists of a simplified, steady-state version of the Kalman Filter. In the equations, T is the sampling interval, $x_{k}$ is the vector with the measured coordinates (longitude and latitude) at time k assumed to be noisy, ${\hat{x}}_{k}$ is the vector estimating the actual coordinates (longitude and latitude) at time k, and ${\hat{v}}_{k}$ is the vector estimating the velocity (over the two components longitude and latitude) at time *k*. The initial state corresponds to the first value of longitude and latitude available while the initial speeds along the two axes are set to zero. At each update, k, the coordinates are estimated using the speed previously computed as follows:(1)$${\hat{x}}_{k}\leftarrow {\hat{x}}_{k-1}+T{\hat{v}}_{k-1}$$

This estimate is independent of new measurements. When new coordinates are made available by the GNSS the coordinates can be updated as follows:(2)$${\hat{x}}_{k}\leftarrow {\hat{x}}_{k}+\alpha \left(x_{k}-{\hat{x}}_{k}\right)$$

Now velocity can be updated as follows:(3)$${\hat{v}}_{k}\leftarrow {\hat{v}}_{k-1}+\beta \left(x_{k}-{\hat{x}}_{k}\right)/T$$

The constants α and β work as smoothing factors. Their optimised value [34] is the following:(4)$$\alpha =1-r^{2}$$
(5)$$\beta =2\left(2-\alpha \right)-4\sqrt{1-\alpha }$$
where:(6)$$\lambda =\frac{\sigma_{w}T^{2}}{\sigma_{n}}$$
(7)$$r=\frac{4+\lambda -\sqrt{8\lambda +\lambda^{2}}}{4}$$

Here $\sigma_{w}^{2}$ is the process variance, which we computed as the confidence interval provided by the phone, while $\sigma_{n}^{2}$ is the noise variance and it was assumed constant, at the value of $3\;{m/s}^{2}$. It is possible to observe that the more λ increases, the more α approaches the value of 1, and the more the algorithm relies on the newly measured coordinates. Conversely, when λ is low, for example, because the confidence interval $\sigma_{n}$ is high, the filter relies more on the coordinates predicted using velocity rather than the newly measured coordinates.

**Kalman 1D:** This Kalman filter can be seen as a generalization of the alpha-beta filter as it also consists of a prediction step based on past state, and a measurement step based on newly gathered information. The equations governing the filter are listed in Figure 4 considering ${\hat{x}}_{k-1}$ and $P_{k-1}$ as initial values of the state and the estimate covariance matrix. A thorough description of the Kalman filter is outside the scope of this paper.

x: State vector;Q: Process noise covariance;A: State transition matrix;B: Control matrix;P: Estimate covariance;R: Measurement covariance;H: Observation matrix;K: Kalman gain;z: Measurements vector.

The choice of how to represent the state and how to compute covariance matrices is what distinguishes the various implementations of the filter. In this one-dimensional case, the state is represented by a single coordinate and its velocity. Therefore, two filters are implemented, one for the longitude and one for the latitude [29]. The process noise covariance matrix (*Q*) is assumed to be $0.25\;{m/s}^{2}$ in [29]. In our case, it is instead derived from the standard deviation of the process (speed and position), which we set as fixed values of 2 m/s and 6 m, respectively. The matrix representing *Q* is based on the following expression:(8)$$Q=\left[\begin{array}{cc} std\_pos\times std\_pos & std\_pos\times std\_speed\\ std\_pos\times std\_speed & std\_speed\times std\_speed \end{array}\right]$$

In [29], a constant value of 1.2 m^2^ is selected as the measurement covariance. In our context, we interpret the confidence interval value associated with each sample as the variance of the measurement. Therefore, we have adopted the square of each sample confidence interval as the value of the scalar (R). As with the alpha-beta, it is possible to observe how a high confidence interval corresponds to a reduced gain, which, in turn, makes the algorithm give higher weight to the prediction rather than the new measurement.

**Kalman 2D:** This algorithm is an extension of the Kalman 1D to two dimensions [30], the equations being the same as in Figure 4. In this version of the filter, longitude and latitude are simultaneously updated through the same set of equations described for the previous method, together with velocity and acceleration. This approach is more reasonable since longitude and latitude are strongly related when a person is walking. Furthermore, in this algorithm, the measurement covariance (R) is computed as the square of the confidence interval associated with each sample. As with the Kalman 1D case, we derive the measurement noise covariance matrix from the fixed values of the standard deviation of the speed and the position and we use the confidence interval of each measured position in the measurement covariance matrix.

**Kalman smoothing:** This implementation is based on the algorithm proposed by Ziegl et al. [22]. Here, the Kalman filter adds a backward recursion to smooth previous samples based on newly collected measures [36]. In [22], the signal is resampled at 1Hz and padded where missing values are found. The initial state of the filter is selected as the first available position sample. The measurement noise covariance (Q) and the observation covariance (*R*, shown in Equation (Equation 9)) were reported based on previous work concerning position tracking through the Kalman filter [37].
(9)$$R=\left[\begin{array}{ccc} 1\times 10^{-8} & 0\times 10^{0} & 0\times 10^{0}\\ 0\times 10^{0} & 1\times 10^{-8} & 0\times 10^{0}\\ 0\times 10^{0} & 0\times 10^{0} & 1\times 10^{4} \end{array}\right]$$

Once the multiple matrixes and initial state are set, each recording is processed. We strove to implement the algorithm as in the original publication, and we verified its fidelity by comparing results from data kindly shared by the authors of the paper.

These six algorithms are compared by considering the difference between the ground truth of the recording and the estimated distance for each algorithm. From the difference, multiple statistics of the obtained error are computed, including mean difference, standard deviation, max difference, Root Mean Square Error (RMSE), and the Bland–Altman Limits of Agreement (LOA).

### 3.4. Data Quality Estimation

We implemented algorithms to assess signal quality during a 6MWT to provide timely user warnings when test reliability was low. To do so, we extracted features from each GNSS recording and analysed them from two perspectives. The first perspective looks at how features relate to the difference between the computed distance using the baseline algorithm and the ground truth (the error), which we will refer to as *error-based* analysis. The second perspective, which we refer to as *user-based* analysis, relates to whether the user performed the test by following the instructions for a valid 6MWT (classified as a conventional test), or did not follow the instructions on purpose to produce unreliable estimations (classified as unconventional).

#### 3.4.1. Features Extraction

It has been observed that an optimal recording scenario involves a trajectory under ideal conditions of satellite visibility, avoiding signal obstruction caused by tall structures like buildings or trees [38]. In addition, path irregularity given from changes in direction may result in less accurate position detection. Based on this, we computed a feature representing changes in direction, which we call “curviness”, corresponding to the measurement of the angle between three consecutive GNSS samples. Additionally to this feature, we considered the value of heading, i.e., the direction of the traveller provided by the GNSS and the delta heading, which corresponds to the difference in consecutive values of heading, thus relating to the angulation of the walk direction. During the filtering phase, it was noticed that occasional inaccurate positions may be included in the signal where the user appears to have walked with a very high speed. For this reason, we included features related to the point-wise speed, obtained from the GNSS signal itself or, if absent, computed between two consecutive samples. As the phone provides an estimation of the confidence interval of the measurement, we included the confidence interval of each GNSS sample as a measure of quality, and the timestamp differences between samples, which can be significant for signal loss. For each of these measures, multiple statistics (see Table 1) were computed, for a total of 52 scalar features. In addition, the following three other summative statistics were added: the percentage of samples of a single test whose GNSS confidence interval is above 15 m, the percentage of samples whose recorded speed is above 4 m/s, and the difference between the distance estimation performed when applying pre-processing filters or not. The last feature can be indicative of the fact that the filters have a big impact on distance estimation and therefore the trace may be affected by noise.

#### 3.4.2. Features Validity

We computed multiple correlation coefficients to validate and identify the association between features and targets for the two considered analyses. The point-biserial correlation [39] and Kolmogorov–Smirnov (KS) statistic were computed in relation to the following binary targets: low error (error < 30 m) and high error (error > 30 m) for the error-based analysis considering only Oxford data, while conventional versus unconventional tests for the user-based analysis considering all available data. The choice of the 30 m threshold is motivated by the Minimal Detectable Change (MDC) of the 6MWD. Among the most conservative values, Chan et al., and Ries et al. [40,41] report a MDC of 28.1 m and 33.5 m, respectively. Given this, we assumed 30 m to be a reasonable choice. Because the MDC can vary across different populations, it is worth mentioning that other studies report higher values of MDC, within 54 m–80 m [42].

#### 3.4.3. Feature Selection and Classification

To select a reduced number of features considered as input of a classification model, we used the Recursive Feature Elimination (RFE) feature selection method with Logistic Regression (LR) as classifier. This method is a wrapper feature selection method that relies on the recursive fit of a model, which, at each iteration, considers fewer features [43]. In particular, a 10-fold cross-validation was repeated five times, considering at each repetition different sets of training and test participants. Five different sets of selected features were obtained and the ones, which were common to at least three of the five sets were considered. Multicollinearity was addressed for both feature sets by observing the characteristics’ Variance Inflation Factor (VIF), which is a metric of the multicollinearity of predictors for a regression analysis task. When using VIF, the accepted threshold can vary for to different applications, where values above 10 usually imply the presence of collinearity between predictors, while for smaller dataset sizes values above 2.5 can also show moderate collinearity [44,45]. In our case, features with VIF higher than 2.5 were linearly combined until there was no such phenomenon.

Once the feature sets were removed from multicollinearity, they were used as input to the following three Machine Learning (ML) classification models: LR, Support Vector Machine (SVM), and Random Forest (RF). These models were selected given their interpretability and low complexity. The classification outcome corresponded to the discrimination between low and high error (threshold at 30 m) for the error-based analysis and to the discrimination between recording types (conventional or unconventional) previously declared by the user for the user-based analysis. The code used to implement the classification models is published with an open-source license at https://github.com/SaraCaramaschi/6mwt_quality, accessed on 15 March 2024.

The models were trained using stratified 5-fold cross-validation and the feature set was normalized by subtracting the mean and dividing by the standard deviation. To evaluate these models, the results from each cross-validation iteration were aggregated and metrics of sensitivity, specificity, F1-score, accuracy, and Area Under the Curve (AUC) were extracted. From the LR classifier, model coefficients were retrieved, odds ratio computed and compared to understand features impact.

## 4. Results

Of the 169 traces, 75 (44%) were produced by 7 pulmonary hypertension patients at Oxford and 94 (56%) by healthy volunteers, 3 at Oxford and 4 at Malmö. Seventy-seven tests (46%) were performed following the guidelines provided by the smartphone app (conventional tests), while 92 (54%) were recorded not following these guidelines (unconventional tests).

The distribution of ground truth 6MWD of the tests is shown in Figure 5. We can see that the Oxford tests are slightly skewed towards distances above 500 m indicating good walking capabilities. On the other side, the tests performed by healthy researchers are generally shorter in distance. This shorter distance was related to the variable duration of those tests between 3 and 6 min, with a focus on generating controlled differences in walking styles during their traces. For this reason, further analysis will consider results in absolute terms for the Oxford set, which included only 6 min-duration tests, while we will include the results in percentage values for all tests from Oxford and Malmö together to obtain an overall relative result. The usage of the datasets across the walked distance estimation and the data quality assessment of this research is further represented for clarity purposes in Figure 6.

### 4.1. Walked Distance Estimation

We applied the pre-processing methods reported in Section 3.2 and computed the walked distance using the algorithms described in Section 3.3. Figure 7 shows the mean absolute percentage error for all the traces, with and without the pre-processing steps (filtering and, where applicable, resampling). It is noticeable that the pre-processing positively impacts all algorithms by significantly reducing the percentage error except for the QSS algorithm, which already incorporates filtering rules in its original design [18]. This indirectly illustrates that the built-in filtering mechanism is effective and its application is relevant.

Table 2 reports the statistics relating to the difference between ground truth 6MWD and estimated 6MWD (the error) after pre-processing for all tests and, in absolute values, using the Oxford dataset. The accuracy is comparable for all algorithms, with the QSS obtaining the lowest mean percentage error with conventional tests (3.04%) and the Kalman 1D for the generalised case and the unconventional tests (8.94% and 13.03%, respectively). Additional information about the performances of the algorithms is provided in the Appendix A, Table A1.

To understand how the shape of the path affects results, Figure 8 reports the absolute percentage error for the tests conducted by healthy volunteers at Malmö with different types of paths considering pre-processed data. The path types “regular” and “stop and go”, which, for the purpose of our analysis, are considered as conventional tests, obtain the lowest median error values, while the recordings including u-turns and circular paths obtain higher median errors. All the algorithms have comparable performances for the same recording type.

Given that no major differences exist in terms of accuracy among algorithms after the data are pre-processed, we will use the baseline algorithm for further analysis, due to the fact that it is easiest to develop and interpret.

### 4.2. Data Quality Estimation

In the following subsection, we go through feature validity, selection and classification results to establish a method to determine data quality. All of the steps consider the two perspectives: the error-based analysis, which has a binary differentiation of low or high error (threshold 30 m), and the user-based analysis, which distinguishes between a conventional or unconventional test. The error-based perspective includes only Oxford 6MWTs and, therefore, the MDC threshold comparison of 30 m is applicable. The user-based perspective considers both tests from Oxford and Malmö given that the test type is not affected by the test duration.

#### 4.2.1. Feature Validity

Feature validity is assessed by computing point-biserial correlation and the KS statistic between each feature and the binary target of the respective analysis (error-based or user-based). These statistics are shown in Figure 9, and numerical values of correlation coefficients are reported in the Appendix A, Table A2.

Table 3 reports the number of tests that fall into the categories of low and high error (the Oxford set) or conventional and unconventional (all tests). The 30 m threshold is not applicable for tests performed in Malmö given that their duration was variable.

#### 4.2.2. Feature Selection

RFE feature selection was performed for both the error-based and the user-based analysis, providing two reduced sets of features to consider as input data for the subsequent classification task. For the error-based analysis, the following six features were selected: crows_median, speed_iqr, curve_iqr, heading_sampen, fs_sampen, and fs_iqr. For the user-based analysis, the following six different features were chosen: quality_sampen, curve_mean, curve_sampen, heading_std, deltaheading_mean, and deltaheading_iqr. Once the two feature sets were decided, multicollinearity was investigated and addressed.

Figure 10 shows the feature correlation maps for the two analyses (error-based and user-based). As expected, high correlation appears between features derived from the same signal characteristic (e.g., fs_sampen and fs_iqr, or deltaheading_mean and deltaheading_iqr).

To remove collinearity, correlated features were combined through simple linear operations. During this process, VIF values were observed iteratively. Within the error-based feature set the median of consecutive distances (crows) and the IQR of the time delta between samples (fs) were summed and the single features dropped, respectively. For the user-based feature set, the delta heading mean and IQR were summed, and curve mean and sample entropy were summed as combined features, while the single components (deltaheading_mean and deltaheading_iqr, curve_mean and curve_sampen) were dropped. The original and final values of VIF are reported in Table 4 for the two analyses.

#### 4.2.3. Classification Results

The results of the three classification algorithms (LR, SVM, and RF) for both the error-based and the user-based perspective are shown in Figure 11. The models’ input corresponds to the set of features derived from RFE selection and reduction of collinearity methods. The Oxford dataset was used for the error-based classification, whereas both datasets were used for the user-based classification. The performances of the binary classification from the error-based analysis reach the highest AUC of 0.83 using the LR algorithm, but the difference from the other models (RF and SVM) is small. The performance of the binary classification for the user-based analysis reaches an AUC of 0.97 for LR and SVM. Table 5 reports the sensitivity, specificity, F1-score, and accuracy of the three models according to the considered analysis. Overall, the models better succeed in distinguishing a conventional test from an unconventional test rather than one with high error from one with low error.

The performances of the LR classifier are promising, given its interpretability and low computational requirements. We report the two corresponding LR equations for error-based analysis, Equation (Equation 10), and for user-based analysis, Equation (Equation 11).
(10)$$P\left(y=High\;error\;6MWT\right)=\frac{1}{\left(1+e^{-}z_{1}\right)}$$
(11)$$P\left(y=Unconventional\;6MWT\right)=\frac{1}{\left(1+e^{-}z_{2}\right)}$$

For error-based analysis, $z_{1}$ corresponds to the following:(12)$$z_{1}=-0.22+0.26X_{1}+0.52X_{2}++0.51X_{3}-0.18X_{4}+0.26X_{5}$$
where the inputs correspond to the following features: $X_{1}$ = speed_iqr, $X_{2}$ = curve_iqr, $X_{3}$ = heading_sampen, $X_{4}$ = fs_sampen, and $X_{5}=$ crows_median + fs_iqr.

For the user-based analysis, $z_{2}$ is as follows:(13)$$z_{2}=0.90+1.03X_{1}+0.81X_{2}-2.65X_{3}+1.54X_{4}$$
where the inputs are as follows: $X_{1}$ = quality_sampen, $X_{2}$ = heading_std, $X_{3}$ = curve_mean + sampen, and $X_{4}$ = deltaheading_mean + iqr.

To observe the actual contribution of each feature, the odds ratios were computed and reported in Figure 12.

## 5. Discussion

### 5.1. Walked Distance Estimation

We investigate the accuracy of a set of algorithms reported in the literature to compute the 6MWD from GNSS data when the test is performed outdoors. All the algorithms work by integrating the distance between consecutive positions received by the GNSS receiver. The ones based on the Kalman filter (alpha-beta, Kalman 1D, Kalman 2D, and the smoothing algorithm from [22]) include a model of the current position, speed, and confidence interval of the user. In contrast, the baseline algorithm and the QSS [18] only make informed decisions about what samples to include in the processing. Other algorithms could be explored, such as deep-learning and data fusion techniques, but since we focus on using consumer technology like smartphones and wearables for the 6-minute walk test, we have opted for simpler techniques.

The results clearly show that all algorithms except the QSS, where pre-processing is already embedded, benefit substantially from resampling and filtering the data to remove samples likely affected by noise. This finding is relevant especially because no previous work investigated the role of pre-processing on accuracy within the context of GNSS-based 6MWT. For example, [22] does not describe any data filtering, but reports an accuracy of 3.7% (for the absolute error), while [20] reports a relative mean error of 8%±32%, though using a correction factor computed by a retrospective analysis of their data. With our dataset, we obtain 20.14% mean percentage absolute error on unfiltered data and 10.35% on filtered data for the same algorithm. If we consider only conventional tests, the absolute error drops to 3.49%, which is comparable to the results provided in the reported papers.

These results show the effect of the shape of the walked path and of the application of filtering techniques to noisy samples and reinforce (a) the need to test algorithms with different walked paths, (b) the importance of providing good instructions to the user, and (c) the need to identify and flag tests that are likely to produce inaccurate results.

Overall, no specific algorithm consistently outperforms the others across different recording types (Table 2), but some distinctions stand out. The QSS algorithm obtains the best accuracy for conventional recordings with percentage LOA of −6.83%, 9.06% and, in absolute terms for Oxford 6MWTs −22.22 m, and 37.31 m. This is comparable to the MDC value of 30 m associated with the 6MWT, and is similar to the value reported in the original paper [18]. Unconventional tests are, however, better tolerated by Kalman-based algorithms, such as the Kalman 1D and Kalman 2D. This can be explained by the fact that the QSS algorithm undersamples the signal, which, in the presence of accentuated curves, approximates the path too much. In the case of unconventional tests, none of the limits of agreement are within even the least conservative MDC value for the 6MWT of 80 m [42]. In other words, if users do not follow instructions, even the currently best algorithms to compensate for this cannot reliably produce clinically relevant walked distance estimations. Instructions that are easily understandable and clear are thus crucial.

Given the similar results obtained by all the considered algorithms after pre-processing, it may be tempting to use the baseline algorithm as it is simplest to implement. However, in conventional tests, this algorithm shows limits of agreements of −44 and 38 m, above the 30 m MDC. According to the Bland-Altman method, the mobile-based version of the 6MWT cannot therefore be considered equivalent to the trundle wheel. It is however relevant to consider that the criterion for the MDC as a threshold for equivalence is quite conservative, and in cases with a softer criterion, the algorithm could still be deemed acceptable. These include e.g., home monitoring contexts where repeated tests can compensate for lower accuracy [46].

### 5.2. Estimating the Quality of Distance Estimation

Previous work has highlighted the problem of low-quality data coming from GNSS signal, in particular, by Stienen et al. [11] in the context of the 6MWT, and by Paziewski [10] in a general framework of usage of the GNSS signal in smartphone applications. In a study from Van Oirschot et al. [24], the authors considered recordings as low-quality if they fulfilled the following criteria: invalid test duration, invalid calculated distance walked, GNSS confidence interval median >30 m, and GNSS confidence interval standard deviation >100 m. Our approach to detect low-quality data, compared with [24], is driven by data instead of somewhat arbitrary rules. By analysing recordings that produce inaccurate distance estimations, we can investigate which properties of the signal characterise these tests. The aim is to detect tests that are likely to be inaccurate at the time of conducting the tests, so that these can be flagged, discarded, used to ask the patient to repeat the test, and potentially also explain to the patient why the test was inaccurate so that they can learn how to follow the instructions.

We identified a set of features that are correlated with the error and/or to the user-reported test type (conventional or unconventional). For both types of classes, we observed a high correlation for features quantifying changes in the direction of the path (the curviness and the delta heading). This is in accordance with causes of inaccuracy reported in previous studies [10,24] and reflects the fact that users performing unconventional tests accentuated the irregularity of the path. Additional information about the validity of the identified features comes from feature selection. The selected features partially confirm the role of some aspects, such as the curviness, but also add aspects that did not come up as strongly correlated, such as the speed IQR. This is justified by the fact that positions affected by noise induce a more irregular velocity, even after filtering. A fairly constant speed is to be expected during the test, except in situations where the user is required to stop, because of fatigue or external factors. In addition, also the confidence interval complexity (quality_sampen) and the difference in time between two consecutive positions are selected (fs_iqr, fs_sampen).

A selection of features that is representative of data quality should subsequently include a mix of factors that indicate curved paths (e.g., curve_mean or deltaheading_mean), variations in how the paths are curved (e.g., deltaheading_iqr, curve_iqr, or heading_std), path complexity (e.g., heading_sampen, or curve_sampen), variations in measurement confidence intervals (quality_sampen), and distances between the positions (e.g., crows_median). Furthermore, the time difference between the samples seems to affect error-based classification (fs_sampen and fs_iqr), this relates to the sampling frequency of the GNSS point, which can vary along the test and across devices. When assessing potential error, speed, distances and curviness are of particular importance, whereas for detecting if a test was performed according to instructions, variations in curviness and confidence interval are more important.

We trained and tested three simple ML models to perform binary classification, for both the error-based and user-based analysis using Oxford 6MWTs for the former, and all available tests for the latter. The results obtained from the error-based classification are promising. However, we can see that the binary classification performed considering the user-based analysis achieves better performances across all observed metrics, as reported in Table 5. Thus, it is easier for the models to differentiate the tests considering the user adherence to the instructions rather than discriminating lower or higher errors produced by the baseline algorithm. From Figure 12, we can see that features considered in the error-based analysis similarly impact the LR model. While, in the user-based analysis, the sum of delta heading mean and IQR (odds ratio of 4.64) appears to have a major impact on the LR equation. Interestingly, the combined feature of the curve mean and the sample entropy sum seems to not affect the model in a noteworthy way. This could potentially be explained by similar information being provided by the delta heading feature.

### 5.3. Limitations and Future Works

Despite the promising results and insights on the discussed topics, some limitations still apply to our study.

The algorithms to compute the walked distance only consider the GNSS signal as information source. We acknowledge the potential of using data fusions techniques by including inertial measurements in further developments and we aim at sharing more work specifically on this topic in the future.

Researchers faced challenges in verifying how individual patients in Oxford reported if they followed the suggested guidelines or not, as no context was provided. In contrast, the data obtained from experimental settings in Malmö were accurately defined and documented. Additionally, when trying to replicate unconventional tests, the primary aspect under control was the path direction. Our efforts to handle the GNSS signal error involved enclosing the smartphone in aluminium foil; however, this intervention did not yield any changes in the GNSS signal error.

Furthermore, the comparison between error-based and user-based classification might be compromised given the difference in datasets used. The presentation of both sets of results remains pertinent though, affording valuable insights that merit investigation in the context of following developments in this domain.

Finally, future enhancements in the performance of binary classification can be achieved by exploring alternative techniques for feature extraction and modelling, such as incorporating deep learning methods.

## 6. Conclusions

The exploration of remote-based functional capacity assessments has been a subject of investigation in the existing literature. However, the current studies tend to overlook the crucial aspect of data quality, often limiting their focus to idealised scenarios. In our study, we present a novel approach to address this gap by introducing a feature-based method to evaluate the quality of the collected data from these tests. Our research aims to provide valuable feedback on whether a smartphone-based, outdoor 6MWT can be deemed of good quality and reliable or not. This information holds significant potential benefits for both end-users and clinicians. On one side, when the feedback indicates low quality, end-users have the opportunity to repeat the test. On the other side, clinicians receive the recorded test results accompanied by quality assessments, enabling them to assess the reliability of the calculated 6MWD. Moreover, being the LR an interpretable model and provided with its coefficients, its implementation on hardware is feasible and it allows the explanation of why the test is likely to be unreliable (e.g., because of the path, or poor satellite visibility).

In this research, we addressed the following two main aspects: the application of filtering rules together with the comparison of algorithms to compute 6MWD, and the quality estimation of a 6MWT. In addition, we provided logistic regression coefficients and features to distinguish 6MWT based on two different perspectives, thereby enabling the system to raise warnings when data quality is likely to be compromised.

In conclusion, this article provides valuable insights into the field of remote 6MWT and the importance of data quality and reliability. We hope that the methodologies and findings presented here will contribute to the advancement of remote functional capacity assessment, ultimately improving the care and monitoring of patients in various healthcare contexts. With these contributions, we look forward to a future where remote functional capacity assessment becomes an indispensable tool in the healthcare industry, facilitating patient care and outcomes. We acknowledge the need for more research and development in this field, as it holds promising ways to assess patients’ functional capacity.

## Figures and Tables

**Figure 1 sensors-24-02632-f001:**
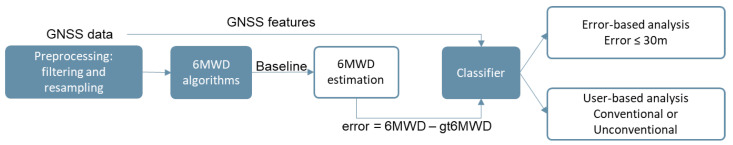
General overview of the research.

**Figure 2 sensors-24-02632-f002:**
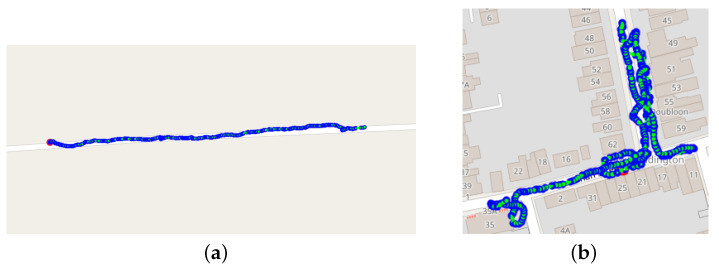
(**a**) Example of conventional 6MWT. (**b**) Example of unconventional 6MWT.

**Figure 3 sensors-24-02632-f003:**
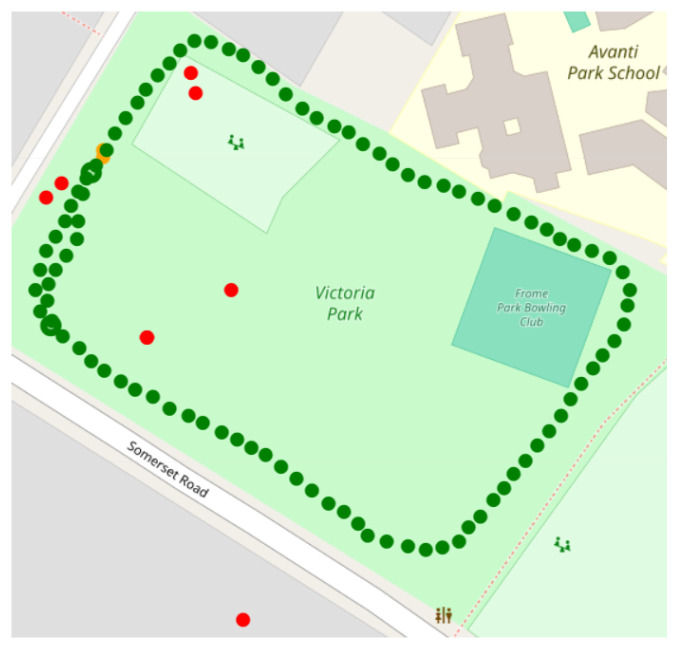
Example of a 6MWT reported on a map. It is visible how some positions are not likely to belong to the actual walked path. In red are samples that are eliminated from the pre-processing filters; in green are samples considered for further analysis.

**Figure 4 sensors-24-02632-f004:**
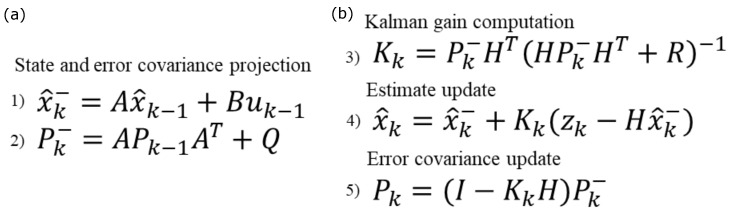
Time update (**a**) and measurement update (**b**) equations.

**Figure 5 sensors-24-02632-f005:**
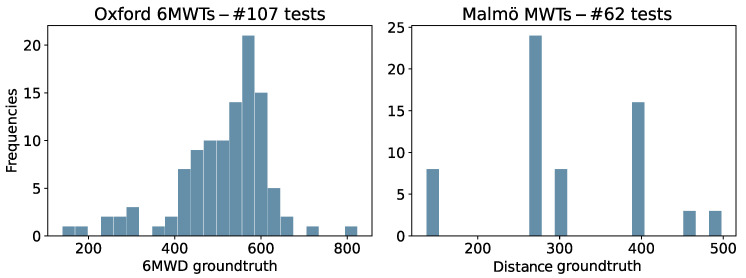
Frequency histogram of the 6MWD reference measurement for Oxford and Malmö tests.

**Figure 6 sensors-24-02632-f006:**
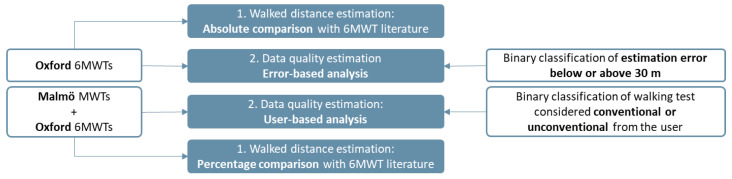
Schema representing the dataset usage across the following two main aspects of this research: distance and data quality estimation.

**Figure 7 sensors-24-02632-f007:**
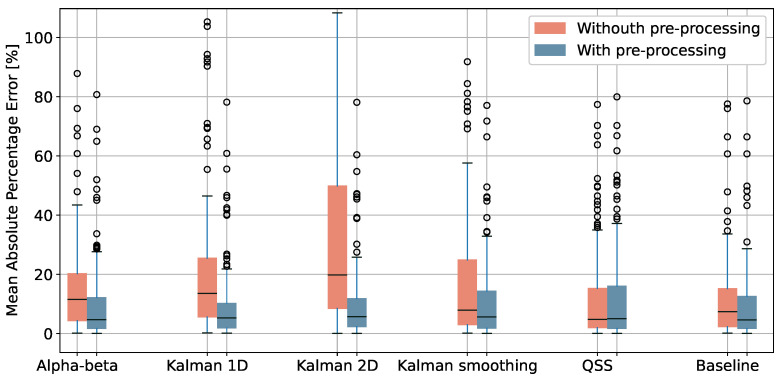
Mean absolute difference, in percentage, between ground truth 6MWD and the one computed by multiple distance estimation algorithms, with and without pre-processing. The error is capped at 100%.

**Figure 8 sensors-24-02632-f008:**
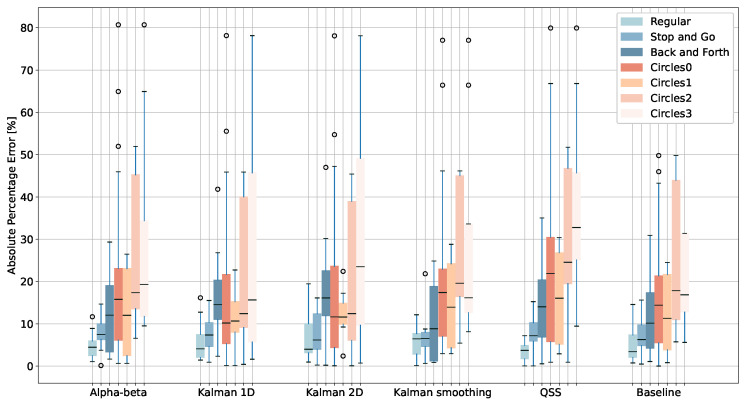
Absolute percentage error for multiple algorithms and recording types of Malmö tests. Data were filtered and resampled except for QSS and baseline algorithms, which were only filtered. The error is capped at 80%.

**Figure 9 sensors-24-02632-f009:**
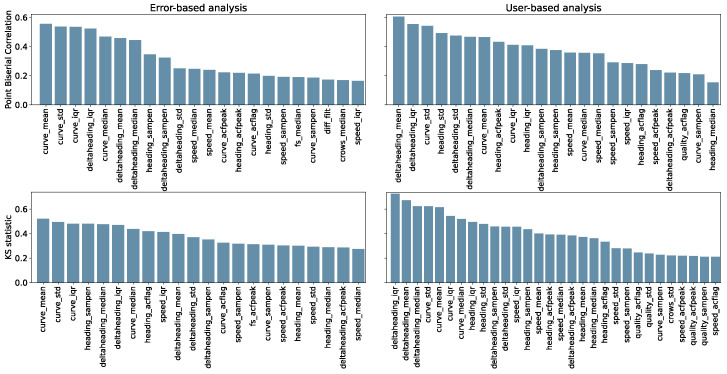
Point-biserial correlation coefficients and KS statistic of the error-based (Oxford only) and user-based (Oxford and Malmö) analyses between features and respective targets. Only statistically significant results (*p* < 0.05) are shown.

**Figure 10 sensors-24-02632-f010:**
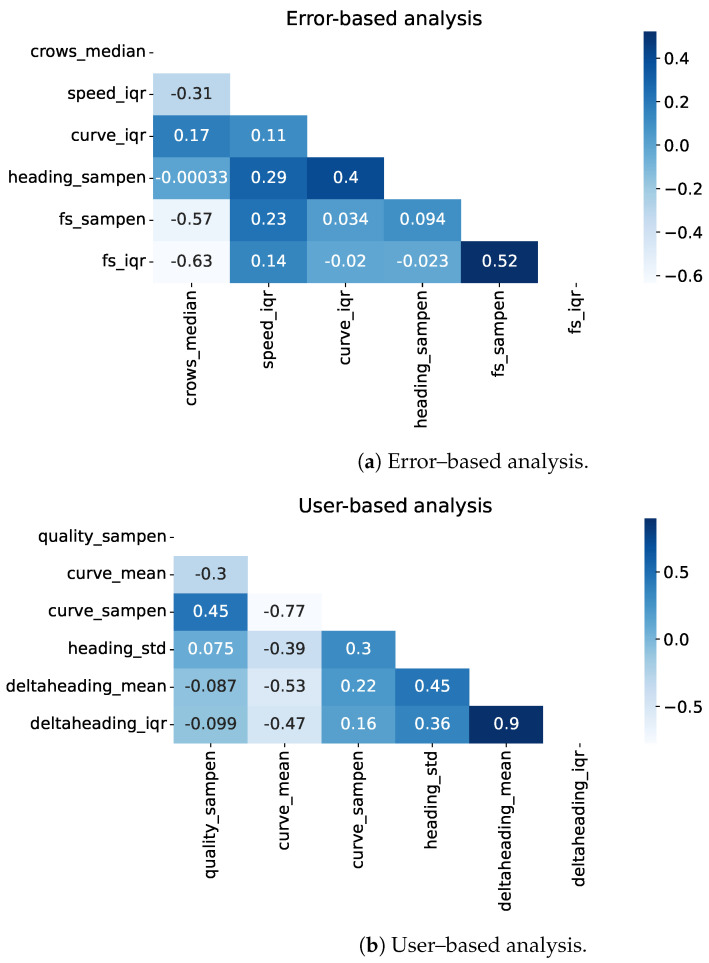
Correlation matrix of the selected features.

**Figure 11 sensors-24-02632-f011:**
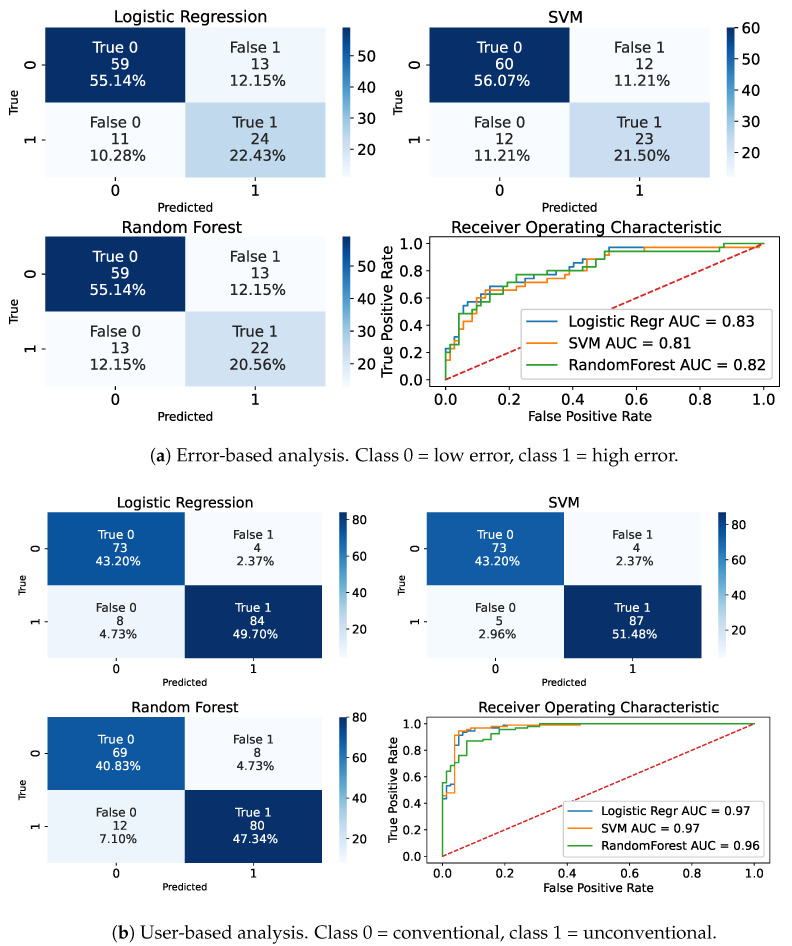
Classification results after feature selection and collinearity reduction.

**Figure 12 sensors-24-02632-f012:**
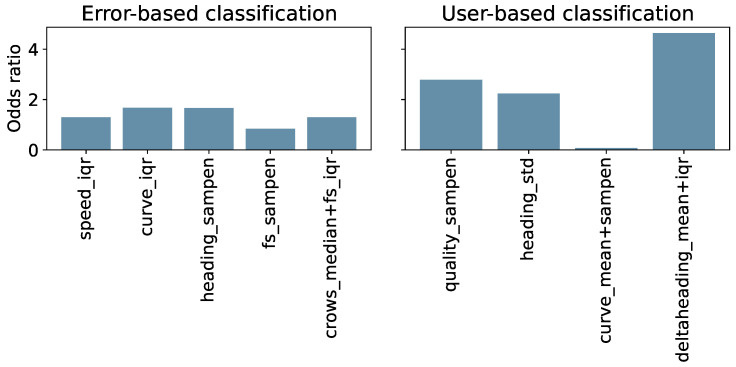
Features odds ratio in error-based and user-based classification of a 6MWT.

**Table 1 sensors-24-02632-t001:** Features computed on the GNSS traces. The left column shows the characteristics of the GNSS signal and the name used in the following sections in reference to specific features. The right column reports the statistics that were computed.

Characteristic of the Measurement	Computed Statistics
Curviness (curve) Point–Point Distance (crows) Speed (speed) Confidence (quality) Heading (heading) Delta Heading (delta heading) Delta Timestamp (fs)	Mean, Median, Standard Deviation, IQR, Autocorrelation 1st Peak Value Autocorrelation 1st Peak Lag Sample entropy
**Additional Features**	
samples (Accuracy > 15) (q_above_tr) %	
samples (Speed > 4) (s_above_tr) %	N/A
Δdistance (unfiltered–filtered)	

**Table 2 sensors-24-02632-t002:** Statistics about the difference between ground truth 6MWD and estimated 6MWD (error) for all algorithms, with pre-processed data. Absolute mean, absolute standard deviation (SD), absolute maximum, and limits of agreement are reported for the error of Oxford 6MWTs and in percentage for all tests. The results are also reported separately for conventional and unconventional test types.

	Oxford Tests #107 [m]			Conventional Oxford #55 [m]			Unconventional Oxford #52 [m]		
	Mean (SD)	Max	LOA	Mean (SD)	Max	LOA	Mean (SD)	Max	LOA
**Alpha-beta**	39.04 (62.59)	401.58	−103.91, 161.94	14.09 (13.57)	62.84	−34.93, 40.86	65.44 (80.68)	401.58	−114.22, 227.35
**Kalman 1D**	**33.6 (52.71)**	384.24	−117.25, 127.04	16.96 (15.35)	64.25	−48.67, 38.95	**51.2 (69.76)**	384.24	−151.75, 182.17
**Kalman 2D**	36.09 (52.57)	380.06	−125.48, 124.49	21.22 (20.28)	100.7	−64.11, 44.03	51.83 (69.07)	380.06	−158.6, 177.79
**Kalman smoothing**	43.5 (68.41)	407.85	−101.3, 178.32	13.37 (12.59)	50.53	−25.25, 40.37	75.36 (86.52)	407.85	−105.03, 247.53
**QSS**	51.06 (81.65)	413.27	−116.16, 211.65	**12.25 (11.73)**	45.89	−22.22, 37.31	92.1 (101.46)	413.27	−111.8, 292.34
**Baseline**	36.5 (58.64)	396.86	−106.95, 150.2	14.47 (14.94)	85	−42.34, 38.92	59.8 (76.05)	396.86	−120.18, 212.8
	**All Tests #169 [%]**			**Conventional All #77 [%]**			**Unconventional All #92 [%]**		
	**Mean (SD)**	**Max**	**LOA**	**Mean (SD)**	**Max**	**LOA**	**Mean (SD)**	**Max**	**LOA**
**Alpha-beta**	9.62 (12.91)	80.7	−23.2, 35.28	3.47 (3.16)	14.68	−8.79, 9.55	14.76 (15.48)	80.7	−25.42, 46.99
**Kalman 1D**	**8.94 (11.84)**	78.14	−28.57, 29.57	4.06 (3.62)	16.15	−11.61, 9.2	**13.03 (14.49)**	78.14	−36.07, 39.94
**Kalman 2D**	9.64 (12.01)	78.08	−31.07, 29.21	4.89 (4.34)	19.47	−14.31, 9.6	13.63 (14.64)	78.08	−38.94, 39.47
**Kalman smoothing**	10.35 (13.29)	77.04	−23.34, 37.01	3.49 (3.46)	21.83	−9.47, 9.8	16.09 (15.56)	77.04	−24.08, 48.92
**QSS**	11.77 (15.48)	79.93	−22.68, 42.68	**3.04 (2.9)**	15.25	−6.83, 9.06	19.07 (17.77)	79.93	−20.55, 55.42
**Baseline**	9.12 (12.28)	78.57	−23.96, 33.21	3.55 (3.44)	15.62	−10.28, 8.92	13.79 (14.81)	78.57	−26.39, 44.52

**Table 3 sensors-24-02632-t003:** Number of tests classified as conventional vs. unconventional or low-error vs. high-error for both datasets. As the data collected in Malmö were of varying duration, it is not possible to fix a threshold for low/high error.

		Error-Based Analysis		
		Low Error	High Error	
User-based analysis	Conventional	43	12	22
	Unconventional	20	35	40
		Oxford		Malmö

**Table 4 sensors-24-02632-t004:** Values of VIF for the original set of selected features obtained through RFE, and the non-collinear set of features for both error-based and user-based analyses.

Error-Based Analysis			
**Original Features**	**VIF**	**Non-Collinear Features**	**VIF**
crows_median	2.22	crows_median+fs_iqr	1.09
speed_iqr	1.24	speed_iqr	1.19
heading_sampen	1.30	heading_sampen	1.29
curve_iqr	1.28	curve_iqr	1.25
fs_sampen	1.63	fs_sampen	1.06
fs_iqr	1.83		
**User-Based Analysis**			
**Original Features**	**VIF**	**Non-Collinear Features**	**VIF**
quality_sampen	1.32	quality_sampen	1.07
curve_mean	3.89	heading_std	1.23
curve_sampen	3.20	curve_mean+sampen	1.36
heading_std	1.34	deltaheading_mean+iqr	1.56
deltaheading_mean	6.04		
deltaheading_iqr	5.17		

**Table 5 sensors-24-02632-t005:** Sensitivity, specificity, F1-score, accuracy, and AUC obtained from the three ML models for the two analyses.

	Error-Based Analysis			User-Based Analysis		
	LR	SVM	RF	LR	SVM	RF
Sensitivity	0.69	0.66	0.63	0.91	0.95	0.87
Specificity	0.82	0.83	0.82	0.95	0.95	0.90
F1-score	0.67	0.66	0.63	0.93	0.95	0.89
Accuracy	0.78	0.78	0.76	0.93	0.95	0.88

## Data Availability

Restrictions apply to the availability of these data.

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
