# Peer review of "Assessing the Effect of Data Quality on Distance Estimation in Smartphone-Based Outdoor 6MWT"

_sensors, 2024, doi:10.3390/s24082632_

Round 1
Reviewer 1 Report
Comments and Suggestions for Authors
As a person from navigation and localization fields, I really don't think the presented approach/study is novel. To be honest I think all the cited and compared methods are far from optimal, e.g. the design of Kalman filter is questionable. Besides the data preprocessing is very common and there is nothing new. I would suggest the authors to read more literatures regarding pedestrian navigation/ localization, and to compare the actual state of the art approaches.
Reviewer 2 Report
Comments and Suggestions for Authors
I have few suggestions for authors to improve this manuscript.
1. Introduction is too short and should be elaborated more.
2. Contributions are missing in this manuscript. Authors are advised to highlight their contributions at the end of Section 1.
3. Section 2 is also short and needs more elaboration and more focus on recently published works.
4. Section 3 is well elaborated. However, I sense that too much data has been messed up. I propose to authors to include a pseudo code of their developed method.
5. Section 4 and 5 are both Okay.
6. Authors are strongly advised to adjust the suggested changes. Hopefully, this work will be a valuable addition for the research community.
Reviewer 3 Report
Comments and Suggestions for Authors
This is an excellent English article that addresses the issue of poor Global Navigation Satellite System (GNSS) signal quality undermining the reliability of test results when conducting the 6-Minute Walk Test (6MWT) using smartphones. The study establishes a framework for assessing the quality and reliability of smartphone-based 6MWT results by applying signal filtering and feature extraction. It provides a framework and ideas for follow-up related research. Carefully reviewing and revising the following suggestions will make the article more comprehensive:
1. Introduction
--- The introduction has several sentences that could be smoother. Please thoroughly review this section, for example:
In line 35: "The primary purpose of Global Navigation Satellite System (GNSS) is primarily to offer precise and immediate global navigation and positioning with respect to a terrestrial reference frame. However, its use in smartphone applications has shown to be reliable but sometimes prone to inaccuracies." Could be modified to:
"The primary purpose of Global Navigation Satellite System (GNSS) is to offer precise and immediate global navigation and positioning with respect to a terrestrial reference frame. However, while its use in smartphone applications has proven to be generally reliable, it can occasionally be susceptible to inaccuracies."
I have provided a sample revision for one sentence, but recommend the author carefully go through the introduction section and revise wording, grammar, and flow as needed to improve fluency throughout.
I congratulate the author on achieving innovative results.
2. Related works
--- In line 104, the author comprehensively analyzed the research on data quality, algorithm adaptability, and other aspects. Besides line 104, the author proposed three research directions. Is it possible to consider data collection in terms of practicality, such as whether the way of wearing a mobile phone in real life (some people are used to putting it in their pockets, some people are used to holding it in their hands, or even putting it in their bags) will affect data collection and data quality.
3. Materials and Methods
---Format issue: The captions of other tables are above the tables, and the caption of Table 1 should also follow the format requirement and be above the table.
4. Results
---The equations in line 486 and line 489 do not have equation numbers, please add them in order.
--- In line 412, the reason why the percentage error of the QSS algorithm did not change much after preprocessing was not explained. It is suggested to add some analysis and explanation from the following perspective: because the QSS algorithm itself has embedded similar filtering rules, which also indirectly proves that the built-in filtering mechanism of the QSS algorithm is effective, and the effect is comparable to adding similar filtering rules additionally.
--The caption of Figure 9 has an extra period “Figure 9. Correlation matrix of the selected features..”
Comments on the Quality of English Language
--- In line 33: "Outdoor solutions are commonly based on Global Navigation Satellite System (GNSS) data while indoor solutions typically rely on inertial sensors or other modalities such as cameras [8]." I would revise this to reflect a parallel structure between outdoor and indoor solutions: "Outdoor solutions commonly rely on Global Navigation Satellite System (GNSS) positioning while indoor solutions typically depend on inertial sensors or other modalities such as cameras [8]."
--- 35 lines, pay attention to the use of professional terminology, the navigation field usually does not use “real-time” instead of “immediate”
Round 2
Reviewer 1 Report
Comments and Suggestions for Authors
I understand that the authors are from different field and are working on classic clinical approach. So I let it pass. But as I have suggested in the previous review, please search pedestrian dead reckoning and pedestrian navigation related literatures and get some ideas from different field.
Reviewer 3 Report
Comments and Suggestions for Authors
The authors have done a good job and revised all of my concerns. I think it suitable to publish. I congratulate the authors on achieving innovative results.